# Overexpression of Extradomain-B Fibronectin is Associated with Invasion of Breast Cancer Cells

**DOI:** 10.3390/cells9081826

**Published:** 2020-08-03

**Authors:** Amita Vaidya, Helen Wang, Victoria Qian, Hannah Gilmore, Zheng-Rong Lu

**Affiliations:** 1Department of Biomedical Engineering, Case Western Reserve University, Cleveland, OH 44106, USA; amv74@case.edu (A.V.); hcw25@case.edu (H.W.); vqian@berkeley.edu (V.Q.); 2Department of Pathology, University Hospitals of Cleveland, Cleveland, OH 44106, USA; hannah.gilmore@case.edu; 3Case Comprehensive Cancer Center, Case Western Reserve University, Cleveland, OH 44106, USA

**Keywords:** EDB-FN, tumor microenvironment, drug resistance, TGF-β, breast cancer, invasion

## Abstract

Breast tumor heterogeneity is a major impediment to oncotherapy. Cancer cells undergo rapid clonal evolution, thereby acquiring significant growth and invasive advantages. The absence of specific markers of these high-risk populations precludes efficient therapeutic and diagnostic management of the disease. Given the critical function of tumor microenvironment in the oncogenic circuitry, we sought to determine the expression profile of the extracellular matrix oncoprotein, extradomain-B fibronectin (EDB-FN) in invasive breast cancer. Analyses of TCGA/GTEx databases and immunostaining of clinical samples found a significant overexpression of EDB-FN in breast tumors, which correlated with poor overall survival. Significant upregulation of EDB-FN was observed in invasive cell populations generated from relatively less invasive MCF7 and MDA-MB-468 cells by long-term TGF-β treatment and acquired chemoresistance. Treatment of the invasive cell populations with an AKT inhibitor (MK2206-HCl) reduced their invasive potential, with a concomitant decrease in their EDB-FN expression, partly through the phosphoAKT-SRp55 pathway. EDB-FN downregulation, with direct RNAi of EDB-FN or indirectly through RNAi of SRp55, also resulted in reduced motility of the invasive cell populations, validating the correlation between EDB-FN expression and invasion of breast cancer cells. These data establish EDB-FN as a promising molecular marker for non-invasive therapeutic surveillance of aggressive breast cancer.

## 1. Introduction

Breast cancer (BCa) is a devastating disease that accounts for 41,000 deaths each year in the US [1]. Although the survival rate for patients with localized BCa is close to 99%, it declines precipitously in patients with distant metastases and drug resistance [1,2]. A major stumbling block in the clinical management of the disease is tumor heterogeneity, which plays a role in the dynamic nature of BCa progression [3]. Whole genome sequencing and profiling studies have demonstrated that breast tumors of the same histological subtype exhibit distinct molecular portraits and discrete trajectories in individual BCa patients at different stages [3,4,5]. Stochastic mutations, genome instability, and clonal evolution arising from selective pressures from genetic, epigenetic, environmental, and therapeutic stimuli result in the emergence of high-risk tumor populations with significant growth and invasive advantages [3,6]. This extensive spatial and temporal diversity within primary and metastasized tumors directly influences diagnostic, therapeutic, and prognostic outcomes [6]. In the absence of markers specific to the metastatic and invasive properties of tumors, current imaging modalities including MRI, PET, and CT are limited in their ability to detect and differentiate between low-risk and high-risk tumors [7]. These facts underscore the need for the discovery and characterization of suitable molecular markers that can facilitate non-invasive detection, risk-stratification, active surveillance of breast neoplasms, and timely assessment of therapeutic response, despite their dynamic nature.

The tumor extracellular matrix (ECM) plays a critical role in all aspects of tumor progression, by relaying oncogenic signals between the tumor cells and the tumor microenvironment (TME) and by supporting growth, apoptotic escape, migration, inflammation, and immune evasion [8]. Fibronectin (FN1), an integral component of normal and tumor ECM, is an essential glycoprotein that regulates adhesion, motility, growth, and development [9]. Its oncofetal alternative splice variant called extradomain-B fibronectin (EDB-FN), however, is known to be overexpressed during malignant transformation, and is generally absent from healthy adult tissues [10,11]. The overexpression of oncofetal fibronectin is correlated with histological grade in mammary tumors [12] and with poor survival in oral carcinoma patients [13]. Multiple lines of evidence show that EDB-FN is associated with epithelial-to-mesenchymal transition (EMT), cancer cell stemness, proliferation, angiogenesis, and metastasis, all of which reflect tumor aggressiveness [14,15,16,17,18]. Clinical studies demonstrate the presence of EDB-FN in patients with lung, brain, colorectal, ovarian, and thyroid cancers [19,20,21], suggesting its potential role as a marker for multiple neoplasms.

An added layer of complication is that even among the same cancer type, EDB-FN expression profiles are distinct and specific to the molecular and functional characteristics of the cells or tissues. For example, using an EDB-FN-specific peptide probe, ZD2-Cy5.5, we previously showed that invasive cancer cell lines, e.g., PC3 (prostate) and MDA-MB-231 (hormone receptor-negative breast cancer), are EDB-FN-rich, while the less invasive cancer cell lines, e.g., LNCaP (prostate) and MCF7 (hormone receptor-positive breast) exhibit significantly lower EDB-FN levels [18,22,23,24]. This differential expression of EDB-FN was exploited for differentially diagnosing invasive prostate and breast cancer tumors from the non-invasive ones using EDB-FN-targeted MRI contrast agents [23,24,25]. Other independent groups have also used EDB-FN as a molecular marker for targeted imaging and therapeutic delivery for various types of cancers [26,27,28].

Given the high degree of tumor plasticity, it is evident that different selective pressures, environmental and experimental stimuli will bring about distinct changes in the TME and EDB-FN expression, which would in turn influence the clinical outcomes of EDB-FN-targeted imaging and therapeutic interventions. Here, we sought to determine the changes in EDB-FN expression patterns following application of two different selective pressures on non-invasive, low-EDB-FN-expressing breast cancer cells and their consequent evolution into invasive high-risk populations. To this end, significant survival advantage was conferred on two breast cancer cell lines, MCF7 and MDA-MB-468, by treating them with the cytokine TGF-β and chemotherapeutic drugs to induce stochastic alternations and clonal evolution. The resulting populations were also treated with a pan-AKT inhibitor to further assess for changes in EDB-FN levels in correlation with the response of the high-risk cells to targeted therapy.

## 2. Materials and Methods

### 2.1. Cell lines and Culture

MCF7, MDA-MB-231, BT549, and Hs578T cells were purchased from ATCC (Manassas, VA, USA). MCF7-DR cells (resistant to 500 nM Palbociclib), MDA-MB-468, and MDA-MB-468-DR (resistant to 100 nM Paclitaxel) cells were a kind gift from Dr. Ruth Keri (CWRU, Cleveland, OH, USA). MCF7-TGF-β and MDA-MB-468-TGF-β cells were obtained by treating the parent lines with 5 ng/mL TGF-β (RnD Systems, Minneapolis, MN, USA) for at least 7-10 days. The breast cancer lines were cultured in Dulbecco’s Modified Eagle’s Medium (DMEM) supplemented with 10% fetal bovine serum (FBS), and 1% Penicillin/Streptomycin (P/S). MCF7, MCF7-DR, and Hs578T cells were additionally supplemented with 0.01 mg/mL human insulin (Sigma-Aldrich, St. Louis, MO, USA). All the cells were grown at 37 °C and 5% CO2. The cell lines were tested for the absence of mycoplasma using the MycoAlert^TM^ Mycoplasma Detection Kit (Lonza, Allendale, NJ, USA). Cell lines were also authenticated by Genetica DNA Laboratories (Burlington, NC, USA).

### 2.2. MK2206-HCl Treatment

The invasive TGF-β-treated and drug-resistant MCF7 and MDA-MB-468 populations were treated with MK2206-HCl, a pan-AKT inhibitor [29], purchased from SelleckChem (Boston, MA, USA). For this treatment, 8 × 10^5^ cells were plated on 6-well plates. After 24 h of attachment, the cells were treated with MK2206-HCl for 2 days (2 µM dose for MCF7 cells and 4 µM dose for MDA-MB-468 cells). Cells treated with equivalent volume of DMSO were used as controls. After treatment, the cells were counted and an equal number of cells was plated on Matrigel and in transwell inserts for the invasion and 3D growth assays, as described in the relevant sections. The treated and non-treated cells were similarly counted and harvested for protein and RNA extraction for western blotting and qRT-PCR, respectively.

### 2.3. Gene Data Analyses

Gene expression analysis and overall survival data were derived from the web server GEPIA2 [30], which provided breast tumor-normal comparison and Kaplan-Meier curve of the EDB-FN transcript (ENST00000432072.6) from TCGA (tumor and normal) and GTEx (normal) databases (1084 BRCA and 291 normal tissue samples). The expression data are first log_2_(TPM+1) transformed for differential analysis and the log_2_FC is defined as median(Tumor)-median(Normal). GEPIA2 uses Log-rank test, or the Mantel–Cox test, for hypothesis test for survival analysis, Cox PH Model for hazards ratio calculation, and ANOVA or LIMMA for differential gene expression analysis.

### 2.4. Immunohistochemistry of Human Tissue Specimens

Human tissue specimens (breast primary tumors, normal adjacent tissue, and metastases in lung, lymph node, and brain) were obtained from the Human Tissue Procurement Facility at Case Western Reserve University (CWRU). All the samples were de-identified and de-classified. Tissue sectioning and IHC services were provided by the Tissue Resources Core Facility of the Comprehensive Cancer Center of CWRU and University Hospitals of Cleveland (grant P30 CA43703). IHC for EDB-FN was performed using 1:100 dilution of anti-EDB-FN antibody G4 clone (Absolute Antibody, UK) after a short antigen retrieval step (30 s at 125 °C in citrate buffer). The slides were reviewed by a certified pathologist. The images were acquired on Bx61VS slide scanner microscope (Olympus, Waltham, MA, USA) with 40X objective lens and were processed in OlyVIA software.

### 2.5. qRT-PCR

Total RNA was extracted from cells using the RNeasy Plus Mini Kit (Qiagen, Germantown, MD, USA), according to manufacturer’s instructions. Reverse transcription was performed using the miScript II RT Kit (Qiagen) and qPCR was performed using the SyBr Green PCR Master Mix (Applied Biosystems, CA, USA). Gene expression was analyzed by the 2^−ΔΔCt^ method with 18S and β-actin levels as the control. The following primer sequences were used: EDB-FN: Fwd 5′-CCGCTAAACTCTTCCACCATTA-3′ and Rev 5′-AGCCCTGTGACTGTGTAGTA-3′; 18S: Fwd 5′-TCAAGAACGAAAGTCGGAGG-3′ and Rev 5′-GGACATCTAAGGGCATC ACA-3′; β-actin Fwd 5′- GTTGTCGACGACGAGCG-3′ and Rev 5′-AGCACAGAGCCTCGC CTTT-3′.

### 2.6. Tumor Spheroid Growth in 3D culture and ZD2-Cy5.5 Staining

For the Matrigel growth assay, 5 × 10^5^ breast cancer cells were suspended in 5% Matrigel-containing media and plated on a thick layer of Corning^TM^ Matrigel^TM^ Membrane Matrix (Corning, NY, USA). The ability of the cells to form tumor spheroids in the 3D Matrix was monitored and photographed for up to 5 days using the Moticam T2 camera with 10× objective lens. After 2–4 days, the cells were stained with Hoechst-33342 (5 µg/mL) and ZD2-Cy5.5 (100 nM) for 30 min at 37 °C. After three washes of PBS, fresh media were added and the cells were imaged on Olympus FV1000 confocal laser scanning microscope (10× objective lens) to obtain Z-stack images. Image analysis was done in FIJI (FIJI Is Just ImageJ) software and ZD2 peptide binding to EDB-FN was quantified as the ratio of the pixel intensities of ZD2-Cy5.5 to that of Hoechst-33342.

### 2.7. Western Blotting

Total cellular protein was extracted from 2D cultures as previously described [31]. Protein concentration was determined by Lowry assay and an equal concentration of protein extracts (40 µg) was loaded on SDS-PAGE gels, transferred onto a nitrocellulose membrane and immunoblotted with primary antibodies overnight. Anti-EDB-FN antibody (G4 clone) was used at 1:1000 dilution. The following primary antibodies (1:1000 dilution) were purchased from Cell Signaling Technology (Danvers, MA): anti-E-cadherin (Cat#3195), anti-Slug (Cat#9585), anti-phospho-T308-AKT (Cat#13038), anti-phospho-S473-AKT (Cat#4060), anti-pan-AKT (Cat#4691), anti-MDR1 (Cat#12683S); and anti-Histone H3 (Cat#4499) and anti-β-actin (Cat#4970) as loading controls. The anti-Phosphoepitope SR proteins (Cat#MABE50; clone 1H4) and anti-SRp40 (Cat#06-1365) antibodies were purchased from Millipore Sigma (Temecula, CA, USA) and used at 1:500 dilution. Anti-N-Cadherin antibody (Cat#76057) was purchased from Abcam (Cambridge, MA, USA) and used at 1:500 dilution. The background-adjusted pixel intensities of test proteins were normalized with those of actin controls in FIJI, and the levels were expressed as ratio of treated over non-treated cells near the respective lanes.

### 2.8. Transwell Assay

For the invasion assay, breast cancer cells were starved in serum-depleted media overnight. The next day, 1–2 × 10^5^ cells were plated in transwell inserts (VWR, Radnor, PA, USA) coated with 0.3 mg/mL Matrigel^TM^ Membrane Matrix. After 1–2 days, the inserts were swabbed with Q-tips to remove the plated cells. The invading cells on the bottom of the inserts were fixed with 4% paraformaldehyde followed by staining with 0.1% crystal violet for 20 min. Excess stain was washed under tap water and images of the purple migrated cells were taken using the Moticam T2 camera with 10× objective lens.

### 2.9. Gene Knockdown Assays

ECO/siRNA nanoparticles were formulated as previously described [32]. Briefly, the amino lipid ECO (5 mM stock in ethanol) was mixed with siLuciferase (as negative control (NC)) or siEDB-FN or siSRSF6 at a final siRNA concentration of 100 nM and N/P = 10 for 30 min to enable self-assembly formation of ECO/siNC, ECO/siEDB, or ECO/siSRSF6 nanoparticles, respectively. For transfections, the nanoparticle formulation was mixed with culture media and added on to the cells. After 24–48 h, the cells were harvested for transwell assays or added onto Matrigel for staining with ZD2-Cy5.5, as described above. The siRNA duplexes: siLuc [sense 5′- CCU ACG CCG AGU ACU UCG AdTdT-3′ and antisense 5′- GGA UGC GGC UCA UGA AGC UdTdT-3′], siEDB-FN [sense 5′- GCA UCG GCC UGA GGU GGA CdTdT-3′ and antisense 5′- GUC CAC CUC AGG CCG AUG CdTdT-3′], and siSRSF6 [sense 5′- CAA AUG AGG GUG UAA UUG AdTdT-3′ and antisense 5′- UCA AUU ACA CCC UCA UUU GdTdT 3′], were purchased from Dharmacon (Lafayette, CO, USA).

### 2.10. Statistical Analyses

All the experiments were independently performed in triplicates (*n* = 3), unless otherwise stated. Bar graphs are represented as mean ± s.e.m. Statistical analysis was performed using Graphpad Prism version 7.03. Data between two groups were compared using unpaired Student’s *t*-test and Mann–Whitney U test. Data between more than two groups were analyzed using 1-way ANOVA or Kruskal–Wallis test, as indicated. *p* < 0.05 was considered to be statistically significant.

## 3. Results

### 3.1. EDB-FN Is Significantly Elevated in Breast Cancer

As a critical ECM component, FN1 is overexpressed in multiple cancer types [11]. Here, the expression of its oncofetal isoform EDB-FN (transcript ID: ENST00000432072.6) in 1084 breast tumor and 291 normal breast samples from TCGA and GTEx databases was evaluated. Differential EDB-FN expression analysis and survival correlation data were derived from the web server GEPIA2 [30]. As shown in Figure 1A, breast tumors demonstrated significant overexpression of EDB-FN (median 36.319 transcripts per million (TPM)), compared to normal breast tissues (median 2.410 TPM), translating to an extremely significant log_2_ fold increase of 3.452 (adj *p* value = 6.57 × 10^−69^) in breast cancer over normal tissues. Furthermore, higher expression of EDB-FN was found to significantly correlate with poor overall survival (Figure 1B) in the breast cancer patients, with a hazard ratio of 1.9 [p(HR)=0.022], demonstrating its prognostic value. To gain insight into the location and patterns of EDB-FN expression in clinical samples, breast tumor specimens and normal adjacent tissues were stained with EDB-FN-specific G4 antibody. As shown in Figure 1C, EDB-FN was abundantly expressed in cancer cell-associated fibroblasts (green), stroma and stromal fibroblasts (red), as well as the mitotic tumor cells (purple) of breast cancer. Further, the expression of EDB-FN was significantly higher in breast tumor tissues compared to the adjacent tissues (Figure 1D). Besides primary tissues, EDB-FN was also highly expressed in lymph node, lung, and brain metastases (Figure 1E), indicating that the oncofetal EDB-FN isoform is highly expressed in malignant breast phenotypes.

We then tested the endogenous EDB-FN expression profile across a panel of cell lines representing the multiple molecular subtypes of breast cancer in correlation with their invasive properties [33,34]. Invasion was measured using standard Matrigel-coated transwell assay and EDB-FN expression in 3D-cultured cells was analyzed using fluorescent-labeled EDB-FN-specific peptide ZD2-Cy5.5 [22]. As shown in Figure 1F, the hormone receptor-positive (HR^+^) MCF7 cells showed the lowest invasion compared to the triple-negative cancer lines, MDA-MB-468, BT549, MDA-MB-231, and Hs578T. In 3D culture, the least invasive MCF7 cells showed the lowest intensity of ZD2-Cy5.5 staining, indicating very low expression of EDB-FN, followed by the more invasive MDA-MB-468 cells (Figure 1G). Finally, the most invasive BT549, MDA-MB-231, and Hs578T cells showed even higher ZD2-Cy5.5 binding, suggesting a direct correlation between EDB-FN expression and cancer cell invasiveness. These results were corroborated in Figure 1H, which showed over a three-fold increase in Hoechst-normalized ZD2-Cy5.5 pixel intensities in the three most invasive breast cancer lines, over MCF7. Further validation was performed by analyzing EDB-FN mRNA expression by qRT-PCR. As shown in Figure 1I, the least invasive MCF7 cells showed the lowest expression of EDB-FN. The more invasive triple-negative breast cancer lines showed significant upregulation of EDB-FN levels, with approximately 9–10-fold increase in MDA-MB-468 and BT549, 14-fold increase in MDA-MB-231 cells, and over 700-fold increase in Hs578T cells. Taken together, these results demonstrate the potential of EDB-FN overexpression as a molecular marker of breast cancer.

### 3.2. Morphological, Functional, and Molecular Changes in 2D- and 3D-Cultured Breast Cancer Cells with TGF-β Treatment and Drug Resistance

To assess the changes in EDB-FN expression levels when breast cancer cells gain significant survival advantages, the two cell lines with the lowest EDB-FN expression and epithelial phenotype, namely MCF7 and MDA-MB-468 cells, were chosen. Two selective pressures were applied: (1) long-term treatment with TGF-β (5 ng/mL) to induce EMT [35] to generate MCF7-TGFβ and MDA-MB-468-TGFβ cells and (2) acquired chemoresistance to Palbociclib, a cyclin-dependent kinase (CDK) inhibitor [36], and to Paclitaxel, an anti-microtubule agent [37], to generate MCF7-DR and MDA-MB-468-DR cells, respectively. Acquisition of drug resistance was confirmed by the significant overexpression of the drug resistance marker P-glycoprotein 1 or multidrug resistance protein (MDR1) in MCF-DR and MDA-MB-468-DR cells (Appendix A). The parent and derivative cell lines were characterized for their morphology, and molecular and functional phenotypes.

As shown in Figure 2A, MCF7 cells demonstrated a typical epithelial morphology in 2D culture. Long-term treatment with TGF-β and development of resistance to Palbociclib resulted in morphological changes to a more mesenchymal phenotype, which was more pronounced in the MCF7-DR cells than in MCF7-TGFβ cells. On the other hand, the MDA-MB-468 cells did not exhibit overt changes in morphology with TGF-β treatment and development of resistance to Paclitaxel. In addition to 2D culture, the cells were grown in Matrigel to facilitate the establishment of a conducive ECM. As shown in Figure 2B, in 3D culture, the low-risk HR^+^ MCF7 cells showed negligible tumor spheroid formation while the more invasive MDA-MB-468 cells showed proliferative network formation. The MCF7-TGFβ and MCF7-DR cells formed tumor spheroids, unlike the parent MCF7 cells, while the MDA-MB-468-TGFβ and MDA-MB-468-DR cells formed similar proliferative networks as their parent counterparts. These results highlight the different properties of each cancer cell type and its distinct response to external mitogenic stimuli.

Next, we analyzed the functional and molecular changes in the TGF-β-treated and drug-resistant breast cancer cells. The treated cell populations were analyzed for their ability to invade through a layer of Matrigel coated in transwell inserts. As shown in Figure 2C, both the TGF-β treatment and drug resistance conferred significant invasive advantage on the MCF7 and MDA-MB-468 cells, rendering them more motile than their parent counterparts, as seen by the increased number of crystal violet-stained migrated cells. TGF-β is a potent inducer of EMT, a critical step towards initiation of metastasis [38]. Similarly, the signaling programs of EMT and drug resistance are intricately related, where EMT-like molecular signature can antagonize chemotherapy in breast cancer [39]. Consequently, the expression of the common EMT markers, N-cadherin (N-cad), E-cadherin (E-cad), and Slug (invasion marker), was tested in the derivative cell lines. As shown in Figure 2D, MCF7-TGF-β cells showed upregulated N-cad and Slug with no change in E-cad levels, while MCF7-DR cells showed upregulation of E-cad and Slug, over MCF7 cells, indicating the gain of invasive properties without loss of E-cad expression in the derivative cells. On the other hand, the MDA-MB-468-TGFβ cells showed no changes in E-cad and N-cad levels, the MDA-MB-468-DR cells showed increased E-cad, N-cad, and Slug expression, compared to the parent cells (Figure 2E). While both the MDA-MB-468-TGFβ and MDA-MB-468-DR cells upregulated the migratory protein Slug, the former did not undergo EMT with TGF-β treatment while the latter possibly gained a hybrid E-M phenotype or a mix of epithelial and mesenchymal cells with drug resistance. Indeed, recent studies have revealed that the partial or hybrid E-M phenotype is attributed to the tumor cell plasticity and is extremely favorable for metastatic dissemination [40,41].

### 3.3. Increased EDB-FN Expression in Breast Cancer Cells with TGF-β Treatment and Drug Resistance

The changes in EDB-FN expression with gain of invasive potential were then determined in the TGF-β-treated and drug-resistant MCF7 and MDA-MB-468 cells. The expression of EDB-FN in 3D-cultured cells was analyzed using fluorescent-labeled EDB-FN-specific peptide ZD2-Cy5.5 [22]. As shown in Figure 3A,B, endogenous EDB-FN expression in MDA-MB-468 cells is higher than that in the MCF7 cells, consistent with that seen in Figure 1G–I, and their invasive ability in Figure 1F. Treatment with TGF-β and acquired drug resistance led to a significant increase in EDB-FN expression in the 3D-cultured MCF7 and MDA-MB-468 cells, reflected in the increased ZD2-Cy5.5 binding in Figure 3A,B and quantified Hoechst-normalized ZD2-Cy5.5 pixel intensities in Figure 3C,D. The peptide binding results were also corroborated by qRT-PCR analysis, which showed over 17-fold and 8.5-fold increase in EDB-FN expression in MCF7-TGFβ and MCF7-DR cells (Figure 3E) and over 3-fold increase in MDA-MB-468-TGFβ and MDA- MB-468-DR cells (Figure 3F), over their respective counterparts. The EDB-FN-specific binding of the ZD2-Cy5.5 probe was confirmed by EDB-FN knockdown experiments in MDA-MB-468-DR cells, where ECO/siEDB nanoparticle treatment abrogated the ZD2-Cy5.5 binding (Appendix A) and silenced EDB-FN mRNA expression by over 70% (Appendix A). These results indicate that, irrespective of the epithelial E-cad or mesenchymal N-cad expression, increased invasion of breast cancer cells results in significant upregulation of EDB-FN. Thus, EDB-FN overexpression is associated with invasive breast cancer cells and with high-risk breast cancer cells that evolve from low-risk ones.

### 3.4. Therapeutic Ablation of Phospho-AKT in Invasive Breast Cancer Cells Decreases Invasion and EDB-FN Expression

Non-invasive therapeutic monitoring of tumor response to oncostatic drugs is crucial to facilitate decision making and timely interventions [6]. To test whether EDB-FN is a therapy-predictive marker and if its expression correlates with changes in the invasive potential of breast cancer cells, the TGF-β-treated and drug-resistant MCF7 and MDA-MB-468 cells were treated with MK2206-HCl, a highly specific pan-AKT inhibitor proven to suppress PI3K/AKT signaling-induced tumor cell proliferation [42]. The PI3K/AKT signaling is a major signal transduction cascade implicated in the progression, metastasis, and drug resistance of multiple cancers [29].

The upregulation of the mitogenic AKT signaling axis in the aggressive TGF-β-treated and drug-resistant MCF7 and MDA-MB-468 cell populations was first confirmed by testing for the levels of phosphorylated AKT (T308 and S473) and total AKT (Figure 4A). Both the phospho-AKT-T308 and phospho-AKT-S473 levels were strongly upregulated in the TGF-β-treated and drug-resistant MCF7 and MDA-MB-468 cells, compared to their respective parent cells. Total AKT was also upregulated in both the cell lines with drug resistance, and, to a lesser extent, with TGF-β treatment. Previous studies have implicated the role of phospho-AKT-SRp40 pathway in the alternative splicing-mediated regulation of EDB-FN expression [43]. To determine the mechanism of EDB-FN upregulation by TGF-β and drug resistance, we examined the expression of phosphorylated SR proteins (SRp55 and SRp40). The more invasive, MCF7-TGFβ, MCF7-DR, MDA-MB-468-TGFβ, and MDA-MB-468-DR cells were found to upregulate the expression of SRp55, while only MCF7-TGFβ, MCF7-DR, and MDA-MB-468-TGFβ upregulated SRp40, compared to their respective parent counterparts. Treatment of the invasive cell derivatives with MK2206-HCl resulted in robust inhibition of phospho-AKT (T308 and S473), as shown in Figure 4B. The upregulation of phosphorylated SRp55 was also diminished in the four cell lines with MK2206-HCl treatment, suggesting that the EDB-FN upregulation in these invasive cells could be controlled, at least in part, through the phospho-AKT-SRp55 signaling pathway (Figure 4B). MK2206-HCl treatment also downregulated SRp40 in the TGF-β-treated and drug-resistant MDA-MB-468 cells, indicating the involvement of more than one splicing protein in EDB exon inclusion.

Functionally, MK2206-HCl-mediated phospho-AKT depletion reduced the invasive potential of the TGFβ-treated and drug-resistant cell derivatives (Figure 4C,D). This was accompanied by a significant decrease in the expression of EDB-FN in the EDB-FN-overexpressing TGF-β-treated and drug-resistant MCF7 and MDA-MB-468 cells, demonstrated by the significant decrease in the mRNA levels (Figure 4E,F) and reduced ZD2-Cy5.5 staining intensities in 3D cultures (Figure 4G–H), compared to the respective DMSO-treated controls. Specifically, in MDA-MB-468-DR cells, although EDB-FN mRNA showed a moderate decrease (about 10–15%), the in situ EDB-FN expressed by 3D spheroids reduced significantly, possibly contributing to the robust decrease in invasion by MK2206-HCl treatment. These results demonstrate a positive association between EDB-FN and invasiveness of breast cancer cell lines, and potential correlation of altered EDB-FN expression with response to therapeutic interventions.

To further explore the link between the SRp55-EDB-FN pathway and invasiveness, the TGF-β-treated and drug-resistant MCF7 and MDA-MB-468 cells were transfected with siEDB-FN- or siSRSF6-bearing nanoparticles to evaluate the loss-of-function effects on the motility of the cells. As shown in Figure 5A, compared to NC (siN), siEDB-FN (siE) and siSRSF6 (siS) treatments resulted in reduced expression of EDB-FN and SRp55, respectively, in the four derivative cell lines. Further, downregulation of SRp55 (siSRSF6) also resulted in downregulation of EDB-FN levels, validating the role of SRp55 in the regulation of EDB-FN expression. Moreover, the reduced levels of both EDB-FN and SRp55 were associated with reduced invasive potential of the TGF-β-treated and drug-resistant MCF7 and MDA-MB-468 cells (Figure 5B,C), suggesting a direct or indirect role of EDB-FN in the altered invasive patterns of breast cancer cells.

## 4. Discussion

Numerous blood biomarkers including CA 15.3, carcinoembryonic antigen (CEA), CA125 and imaging modalities like ultrasound, mammography, MRI, PET, and CT are routinely used to detect primary breast tumor disease and recurrence and to assess therapeutic response [44,45]. However, they are limited in their ability to differentially diagnose and risk-stratify the disease, with high rates of false positive diagnoses [46], underscoring the need for specific markers to accurately detect highly invasive and metastatic breast tumors, and to distinguish them from low-risk indolent ones. Moreover, breast tumors frequently exhibit intrinsic or acquired resistance to chemotherapy and targeted drugs [39]. In the absence of suitable molecular markers, active surveillance and monitoring of the efficacy of chemotherapeutic interventions and timely detection of the emergence of resistant phenotypes forms another obstacle to patient treatment.

To address these concerns, this study investigated the dynamic changes in the ECM oncoprotein EDB-FN in conjunction with the dynamic changes in the invasive potential of breast cancer cells. We found that invasive cells that evolve from low-risk cancer cells overexpress EDB-FN, irrespective of their epithelial or mesenchymal markers; conversely, impeding the invasive abilities of these high-risk cancer cells with a targeted drug abolishes their EDB-FN overexpression, demonstrating a direct correlation between EDB-FN levels and the invasiveness of breast cancer cells. Moreover, to our knowledge, this is the first study to report that EDB-FN is upregulated with development of drug resistance in breast cancer cells.

Between the two different breast cancer lines used in this work, the endogenous EDB-FN level in the least aggressive HR^+^ MCF7 cells is significantly lower than that in the more aggressive HR^−^ MDA-MB-468 cells, despite both lines exhibiting an epithelial phenotype. Induction of drug resistance and long-term TGF-β treatment led to distinct changes in the molecular phenotypes of the two cell lines, possibly through distinct signaling mechanisms. The emergent invasive populations became more aggressive than their parent cells, with a partial E-M phenotype seen in MDA-MB-468-DR cells, and increased phospho-AKT signaling. Indeed, recent studies suggest that the existence of subpopulations of cancer cells along different positions on the E-M spectrum confers profound plasticity on the tumors, promoting their progression, metastasis, and stemness [40]. While we observed increased expression of the pro-invasive protein Slug in both the cell lines with long-term TGF-β treatment and drug resistance, it is likely that each of these cell lines acquired their survival advantages through other known pathways like Hedgehog, NF-κB, PI3K-AKT-mTOR [47], and additional here-to-fore unstudied mechanisms. Although phospho-AKT signaling was robustly upregulated in the cells, it would be of significant future interest to gain in-depth insight into the precise causal factors of their increased motility. Nevertheless, all of the cells with acquired invasiveness presented elevated EDB-FN expression irrespective of the signaling mechanisms. Conversely, when the aggressive cells were treated with targeted therapy, their invasive potential diminished with a concomitant reduction in EDB-FN expression, indicating the role of EDB-FN as a marker for active surveillance and monitoring of therapeutic efficacy. Given that FN1 has several other alternatively spliced isoforms, including EDB/EDA^−^ FN1 and EDA-FN, further extensive studies are required to evaluate the specificity and/or sufficiency of the role of EDB-FN in invasion. In view of the numerous studies demonstrating overexpression of EDB/EDA^−^ FN1 in tumorigenic ECM [11,48], it can be speculated that it may also be upregulated in our TGF-β-treated and drug-resistant breast cancer cells. However, its abundance in healthy tissues renders it an unsuitable tumor-specific marker, and did not comprise the focus of our research. On the other hand, the oncofetal isoform EDA-FN is under investigation as a potential oncomarker, and its role in breast cancer is described elsewhere [49,50].

The precise mechanism of EDB-FN upregulation in invasive cells remains an enigma. At the genetic level, EDB-FN is generated by alternative splicing event, resulting in the inclusion of the EDB exon in the FN1 transcript, a process controlled by SR (Ser- and Arg-rich) proteins of the splicing regulator family [10,51]. Since alternative splicing is indispensable for the formation of the EDB-FN isoform, the participation of the SR proteins in this process is inevitable. However, there is limited research on the underlying mechanism of the preferential and differential inclusion of the EDB exon during neoplastic transformation. Previous studies show that increased tissue stiffness directly upregulates PI3K/AKT-mediated SRp40 phosphorylation, enhancing exon inclusion and EDB-FN secretion by breast cancer cells [43]. In this study, development of drug resistance and TGF-β treatment in MCF7 and MDA-MB-468 cells consistently upregulated SRp55 phosphorylation, in addition to increased phospho-AKT. Knockdown of SRp55, directly using RNAi or indirectly using MK2206-HCl, significantly reduced their EDB-FN expression as well as invasive potential. Given that SRp55 is commonly mutated in breast and colorectal cancers [52], and influences the alternative splicing patterns of several tumor-associated genes like KIT, CD44, and FGFR1 [53], it is not surprising that SRp55 depletion decreased the invasion of breast cancer cells. However, this is the first study to report decreased EDB-FN expression as a consequence of SRp55 depletion. How SRp55, and the other SR proteins, act in conjunction with their antagonistic hnRNPs in the spliceosome [53], to regulate the complex alternative splicing processes in response to various intrinsic and extrinsic stimuli remains to be explored. Additionally, MK2206-HCl treatment showed highly specific knockdown of phospho-AKT and consequent downregulation of SRp40/SRp55 levels in a cell-specific manner. While the MK2206-HCl treatment significantly reduced EDB-FN expression and invasion, it did not completely abrogate them, suggesting the compensatory activation of other mitogenic proteins (like AKT3) [54] or the EDA-FN isoform, which is also involved in tumorigenesis [11]. This study focused on analyzing the phospho-AKT-SR phosphorylation axis in 2D cell cultures. Given the complex composition and molecular signaling in breast malignancies, it would be interesting to evaluate the distinct spatial and temporal changes in EDB-FN expression and function following different drug treatments in 3D cultures.

EDB-FN is overexpressed in multiple types of cancer, including colorectal, oral, bladder, lung, and prostate [13,19,55,56,57]. Originally thought to be secreted only by cancer-associated fibroblasts (CAFs) and endothelial cells, EDB-FN is now known to be abundantly produced by tumor cells, especially invasive tumor cells [11]. EDB-FN is upregulated during embryogenesis, temporally activated during wound healing, tissue repair, and angiogenesis, but mostly absent from healthy adult tissues [10]. Indeed, immunohistochemical analysis in our study showed robust EDB-FN expression in the mitotic breast tumor cells, diffuse staining in the stroma, stromal fibroblasts, and fibroblasts interspersed between cancer cells, in primary tumor specimens as well as metastatic sites, with negligible expression in normal breast tissues, suggesting complex structural and functional roles of EDB-FN in the progression of malignancies. In TCGA/GTEx samples, EDB-FN is significantly overexpressed in breast cancer and is negatively correlated with patient survival. Additionally, by virtue of its extracellular location and ready accessibility, EDB-FN has emerged as an attractive target for designing new diagnostic and therapeutic regimens. Previous studies have already demonstrated the potential of antibody-mediated EDB-FN targeting (using L19, BC-1) for angiogenesis, inflammation, and cancer stem cell therapy [58,59]. EDB-FN-specific peptides, such as ZD2 and APT_EDB_, are advantageous for oncogenic ECM targeting, by virtue of their small size, low immunogenicity, and high tissue penetration ability [22,26,60]. The specificity and superior binding of the ZD2 probe for EDB-FN has direct translational implications. We have successfully demonstrated differential diagnosis of non-invasive and invasive breast and prostate cancer xenografts in mouse models using ZD2-targeted MRI contrast agents [18,23,24,25]. The results of this study open up new avenues for determining the potential of EDB-FN as a marker for molecular imaging-based detection, risk-stratification, active surveillance, and monitoring of breast cancers and tracking their evolution as the disease progresses with and without chemotherapy.

## 5. Conclusions

In summary, this research shows that EDB-FN expression is strongly associated with highly invasive breast cancer and with low-risk cells that evolve into high-risk ones. This correlation holds true despite cancer cell plasticity, and dynamic changes occurring in the invasive properties of breast cancer cells lead to corresponding changes in the EDB-FN expression levels. In addition to demonstrating for the first time that acquired drug resistance upregulates EDB-FN, we also show that this enhanced expression, in part, occurs through the phospho-AKT-SRp55 signaling pathway. These observations indicate that EDB-FN is a promising molecular marker for monitoring the progression of breast cancer, in the context of diagnostic imaging and therapeutic interventions.

## Figures and Tables

**Figure 1 cells-09-01826-f001:**
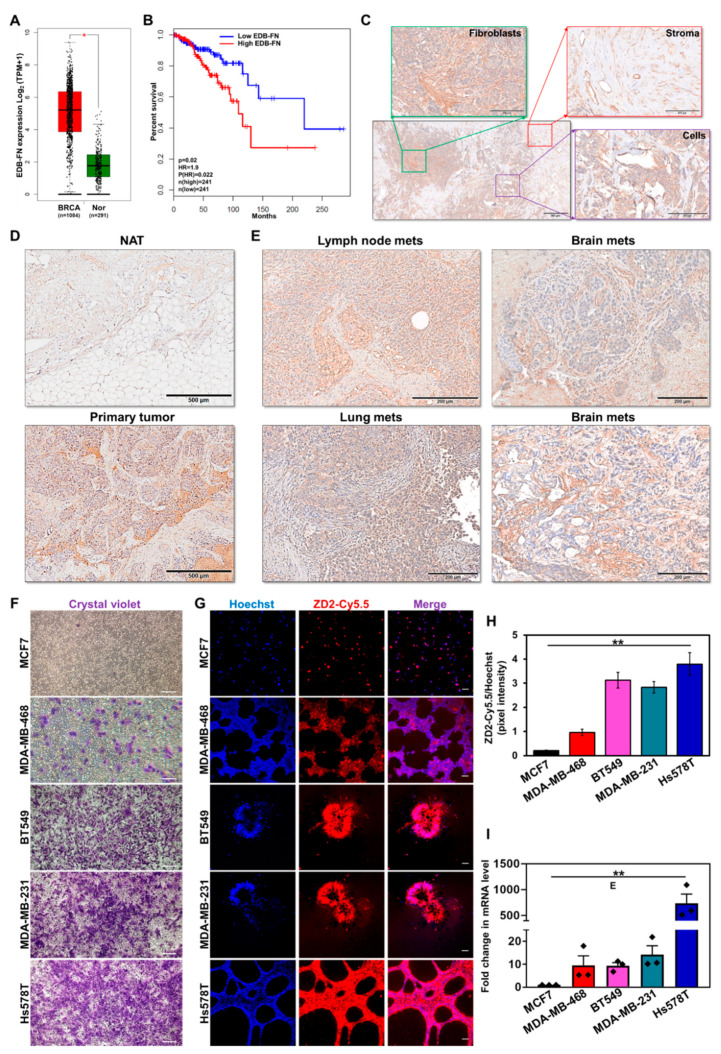
Extradomain-B Fibronectin (EDB-FN) overexpression in invasive breast cancer. (**A**) Differential gene expression analysis performed on patient data from the TCGA and GTEx databases shows significant overexpression of EDB-FN isoform (ENST00000432072.6) in breast tumor samples (BRCA, *n* = 1084) compared to normal breast tissue samples (Nor, *n* = 291). Box plot denotes log_2_ (TPM+1) transcripts per million values and median with interquartile range, * *p* = 6.57 × 10^−69^ using ANOVA. (**B**) Kaplan–Meier curves show overall survival analyses of breast cancer patients with EDB-FN expression (* *p* < 0.05; * *p* for Hazard ratio = 0.022 using Log-rank test, i.e., Mantel-Cox test). (**C**) Immunohistochemical staining pattern of EDB-FN in primary breast tumors showing expression in fibroblasts (green), cancer cells (purple), and tumor stroma (red). Strong EDB-FN staining observed in (**D**) primary breast tumor compared to normal adjacent tissue (NAT), and metastases in (**E**) lymph node, lung, and brains. (**F**) Comparison of the invasive potential of triple-negative MDA-MB-468, BT549, MDA-MB-231, and Hs578T cell lines, with hormone receptor-positive MCF7 line by transwell assay. (**G**) 3D cultures of the cancer cell lines stained with EDB-FN-specific ZD2-Cy5.5 probe show higher EDB-FN expression in the invasive cell lines than in MCF7 cells, demonstrating direct correlation with their invasive pattern. (**H**) Quantification of ZD2 peptide binding to EDB-FN, using FIJI, as the ratio of the pixel intensities of ZD2-Cy5.5 to that of Hoechst. Bars denote mean ± sem (*n* = 3). ** *p* = 0.011 using 1-way ANOVA. (**I**) qRT-PCR analysis exhibits significantly elevated levels of EDB-FN mRNA in the invasive cell lines, compared to less invasive MCF7 cells. 18S expression was used as standard. Bars denote mean ± sem and dots denote 3 technical replicates from 3 independent experiments, ** *p* = 0.0013 using Kruskal–Wallis test.

**Figure 2 cells-09-01826-f002:**
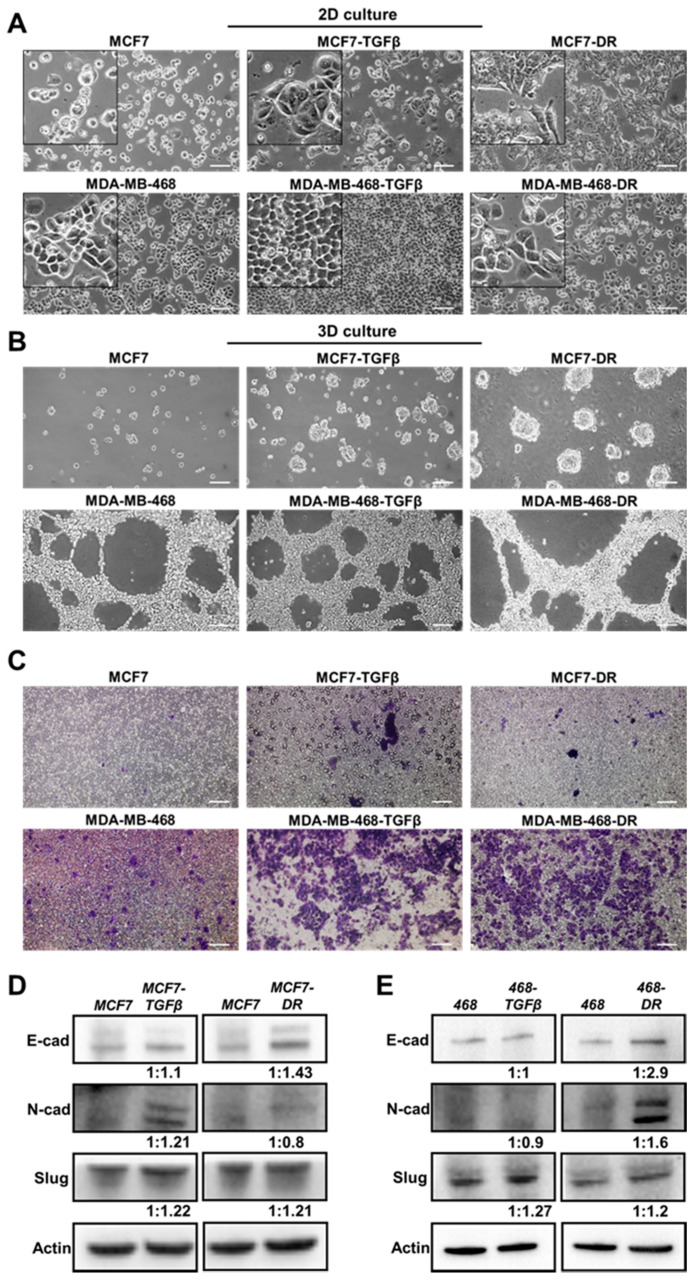
Altered growth and morphology and enhanced invasion in breast cancer cells with TGF-β treatment and drug resistance. MCF7 and MDA-MB-468 cells were cultured in 5 ng/mL TGF-β for 7–15 days to obtain MCF7-TGF-β and MDA-MB-468-TGF-β, cells, respectively. MCF7-DR and MDA-MB-468-DR cells were obtained by inducing resistance to 500 nM Palbociclib and 100 nM Paclitaxel respectively. (**A**) In 2D culture, MCF7-TGFβ and MCF7-DR cells show distinct morphological changes, with a more mesenchymal phenotype, while MDA-MB-468-TGFβ and MDA-MB-468-DR cells do not show visible morphological changes, compared to their parent lines. (**B**) In 3D Matrigel culture, MCF7-TGFβ and MCF7-DR cells show increased ability to form spheroids, while MDA-MB-468-TGFβ and MDA-MB-468-DR cells form similar proliferative networks, compared to their respective parent counterparts. (**C**) Transwell assay shows higher invasive potential of TGF-β-treated and drug-resistant MCF7 and MDA-MB-468 cells, evidenced by the increase in number of crystal violet-stained migrated cells. Scale bar = 100 µm. Western blot analysis for EMT markers shows protein expression of E-cadherin, N-cadherin, and Slug in (**D**) MCF7-TGFβ and MCF7-DR and (**E**) MDA-MB-468-TGFβ and MDA-MB-468-DR cells, compared to the parent MCF7 and MDA-MB-468 cells, respectively. The western blot band intensities were normalized to those of actin bands in FIJI, and the protein level changes are expressed as ratio of treated over non-treated, below the respective lanes.

**Figure 3 cells-09-01826-f003:**
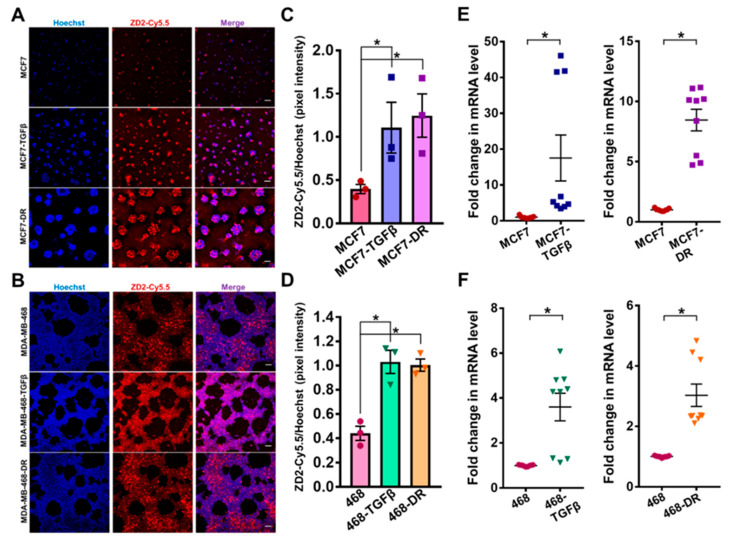
Increased EDB-FN expression in breast cancer cells with TGF-β treatment and drug resistance. ZD2-Cy5.5 staining of 3D cultures of breast cancer cells shows significantly increased EDB-FN expression in (**A**) MCF7-TGF-β and MCF7-DR and (**B**) MDA-MB-468-TGF-β and MDA-MB-468-DR cells, compared to the parent MCF7 and MDA-MB-468 cells, respectively. ZD2 peptide binding to EDB-FN was quantified in FIJI as the ratio of the pixel intensities of ZD2-Cy5.5 to that of Hoechst to show increased EDB-FN level in (**C**) MCF7-TGF-β and MCF7-DR and (**D**) MDA-MB-468-TGF-β and MDA-MB-468-DR cells, compared to the parent MCF7 and MDA-MB-468 cells. Bars denote mean ± sem and dots denote independent replicates. * *p* < 0.05 using Mann–Whitney U test. qRT-PCR analysis shows significantly upregulated mRNA expression of EDB-FN in (**E**) MCF7-TGFβ and MCF7-DR and (**F**) MDA-MB-468-TGFβ and MDA-MB-468-DR cells, compared to the parent MCF7 and MDA-MB-468 cells, respectively. Dots denote 3 technical replicates from 3 independent experiments, with lines at mean ± sem. * *p* < 0.05 using Mann–Whitney U test. Scale bars = 100 µm.

**Figure 4 cells-09-01826-f004:**
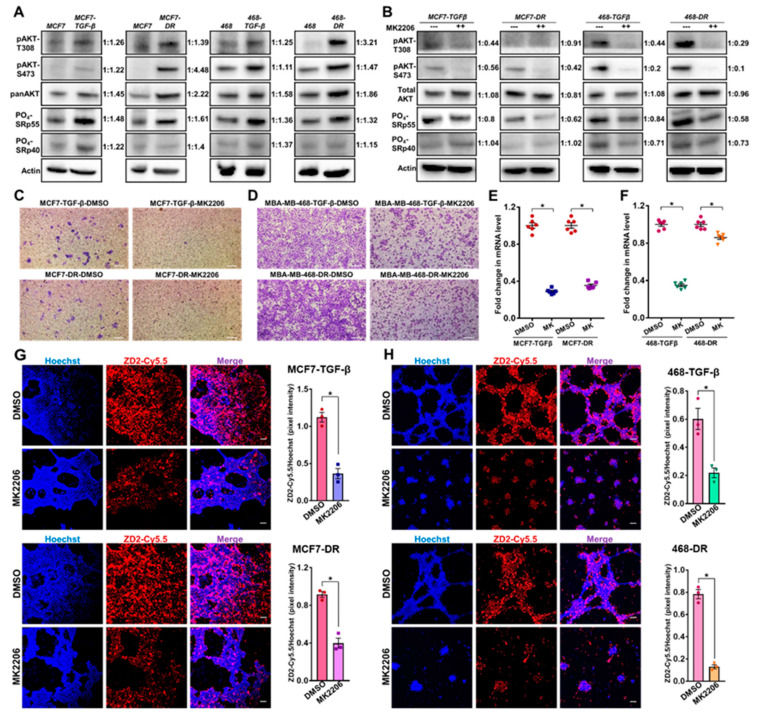
Therapeutic ablation of phospho-AKT in invasive TGF-β-treated and drug-resistant breast cancer cells reduces their invasion and EDB-FN overexpression. (**A**) TGF-β treatment and drug resistance upregulate phospho-AKT signaling and phosphorylation of SRp55 in MCF7 and MDA-MB-468 cells. (**B**) Treatment of cells with AKT inhibitor MK2206-HCl (2 µM for MCF7 and 4 µM for MDA-MB-468 cells) for 2 days results in inhibition of phospho-AKT signaling in the invasive breast cancer cell lines. The SRp55 upregulation is diminished with MK2206-HCl-mediated depletion of phospho-AKT signaling, suggesting a potential role of SRp55 in the inclusion of EDB-FN exon. The western blot band intensities were normalized to those of actin bands in FIJI, and the protein level changes are expressed as ratio of treated over non-treated, next to the respective lanes. Inhibition of phospho-AKT signaling by MK2206-HCl demonstrates reduced invasive potential of the invasive (**C**) MCF7 and (**D**) MDA-MB-468 cell derivatives along with decreased expression of EDB-FN at (**E**) and (**F**) mRNA levels and (**G**) and (**H**) in 3D culture, respectively. Dots denote 2 technical replicates from 3 independent experiments, with lines at mean ± sem. (**G–H**) ZD2 peptide binding to EDB-FN was quantified in FIJI as the ratio of the pixel intensities of ZD2-Cy5.5 to that of Hoechst. Bars denote mean ± sem and dots denote independent replicates. * *p* < 0.05 using Mann–Whitney U test. Scale bars = 100 µm.

**Figure 5 cells-09-01826-f005:**
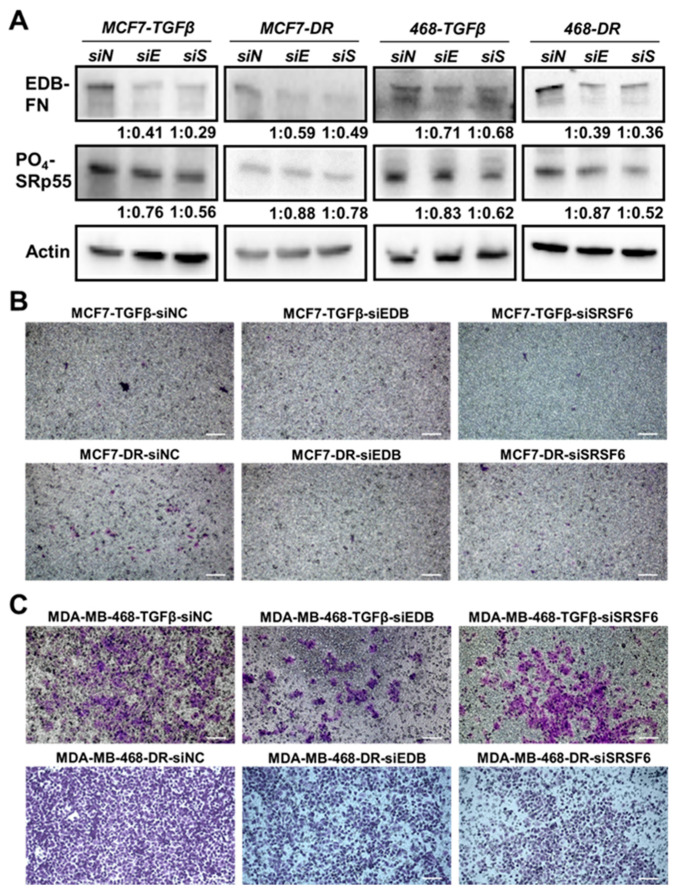
Depletion of EDB-FN and SRp55 in invasive TGF-β-treated and drug-resistant breast cancer cells reduces their invasion. TGF-β-treated and drug-resistant MCF7 and MDA-MB-468 cells were treated with ECO/siNC (siN), ECO/siEDB (siE), and ECO/siSRSF6 (siS) nanoparticles at 100 nM [siRNA] and N/P = 10. (**A**) Knockdown of EDB-FN and SRSF6 results in downregulation of EDB-FN expression and SRp55 levels with a concomitant reduction in the invasive potential of the invasive TGF-β-treated and drug-resistant (**B**) MCF7 and (**C**) MDA-MB-468 cells. The western blot band intensities were normalized to those of actin bands in FIJI, and the protein level changes are expressed as ratio of siEDB or siSRSF6 over siNC, below the respective lanes. Scale bar = 100 µm.

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
