# Peer review of "Overexpression of Extradomain-B Fibronectin is Associated with Invasion of Breast Cancer Cells"

_cells, 2020, doi:10.3390/cells9081826_

Round 1
Reviewer 1 Report
There are two major problems remaining in the revised manuscript concerning me.
- The authors did not respond to the problem of EDB-FN specificity. Is EDB in FN truly that essential for the invasiveness of breast cancer cells? How about overexpression of FN without EDB? To exhibit the specificity of EDB FN, they should have shown that the overexpression of cellular FN only containing EDA, but not EDB, is not associated with invasion of breast cancer cells.
- The manuscript was attempted to mainly associate overexpression of EDB FN with invasion of breast cancer cells. The purpose of performing Fig. 2 by treating cells with TGFb and generating drug resistant cells was to produce breast cancer cells with more invasive properties (Fig. 2A-2C) for the association with EDB-FN. I still don't think Fig. 2D and 2E are necessary for that purpose and the results may even make the manuscript distracting (because the manuscript is not focused on EMT issues). Based on Fig. 2D and 2E, the authors claimed that both the TGFb-treated and drug-resistant MCF7/MDA-MB-468 cells upregulated the migratory protein Slug, whereas the effects on E-cad and N-cad expressions were less consistent, together revealing hybrid E-M phenotype. Such E-M phenotype will become more valuable if they are able to demonstrate that silencing Slug (the only EMT protein expression consistent with invasiveness of cancer cells), but not E-cad or N-cad, can reduce invasion of MCF7/MDA-MB-468 cells.
Author Response
We greatly appreciate the constructive criticism and comments of the reviewers to improve the quality of our manuscript. We found that the comments and suggestions are also valuable for our future research work. We have addressed the comments and concerns from the reviewers point-by-point as the following.
Reviewer 1
There are two major problems remaining in the revised manuscript concerning me.
The authors did not respond to the problem of EDB-FN specificity. Is EDB in FN truly that essential for the invasiveness of breast cancer cells? How about overexpression of FN without EDB? To exhibit the specificity of EDB FN, they should have shown that the overexpression of cellular FN only containing EDA, but not EDB, is not associated with invasion of breast cancer cells.
R: Naturally, normal FN will also be overexpressed, and there are numerous reports (e.g. Lin et al, Cells. 20209(1)27). However, given that it is also expressed in normal tissues, FN1 is not a suitable marker for tumor targeting. Hence, we only focused on EDB-FN, because it is generally considered as an oncofetal version of FN1 and has significantly high expression in aggressive tumors and low expression in healthy tissues. EDB-FN can be used as a viable target for therapy and imaging as shown in Han et al, Nat Comm 2017, 8:692; Han et al., Magn Reson Med 2018, 79(6):3135-43 and other independent reports in the literature. There are also reports of EDA-FN being overexpressed in breast cancer (Kwon et al., J Cellular Physiology 2019, https://doi.org/10.1002/jcp.29326), and is studied as a marker for the same reason. In this study, the focus in this manuscript is not about the specificity of EDB-FN or that it is essential for invasiveness, rather it is about it’s potential association with invasiveness when different pressures are applied to 2 breast cancer cell lines.
The manuscript was attempted to mainly associate overexpression of EDB FN with invasion of breast cancer cells. The purpose of performing Fig. 2 by treating cells with TGFb and generating drug resistant cells was to produce breast cancer cells with more invasive properties (Fig. 2A-2C) for the association with EDB-FN. I still don't think Fig. 2D and 2E are necessary for that purpose and the results may even make the manuscript distracting (because the manuscript is not focused on EMT issues). Based on Fig. 2D and 2E, the authors claimed that both the TGFb-treated and drug-resistant MCF7/MDA-MB-468 cells upregulated the migratory protein Slug, whereas the effects on E-cad and N-cad expressions were less consistent, together revealing hybrid E-M phenotype. Such E-M phenotype will become more valuable if they are able to demonstrate that silencing Slug (the only EMT protein expression consistent with invasiveness of cancer cells), but not E-cad or N-cad, can reduce invasion of MCF7/MDA-MB-468 cells.
R: As the reviewer rightly said, the manuscript is not focused on EMT. E-cadherin and N-cadherin were tested since they are most common markers of EMT. Besides slug, it is very likely that multiple pathways like Hedgehog, NF-κB, etc. could be involved (as reviewed by Delou et al., Highlights in Resistance Mechanism Pathways for Combination Therapy. Cells. 2019 Sept 8(9):1013) in the invasiveness of the derivative cell lines and so, silencing Slug may not be sufficient to explain invasion. Consequently, figuring out which proteins cause this invasion would complicate the interpretation and take away from the focus of EDB-FN and invasion. Nevertheless, we will perform the experiments suggested by the reviewers in the future studies.

Reviewer 2 Report
Vaidya et al. found that upregulation of EDB-FN in aggressive breast cancers. The authors used MCF-7 and MDA-MB-468 cells. TGFbeta-treated cells and chemoresistant cells showed higher EDB-FN expression than parental cells. Moreover, in these cells, AKT inhibitor treatment downregulated EDB-FN expression and reduced the invasive ability. SRSF6 knockdown also reduced EDB-FN expression and the invasive ability. The manuscript is well organized. The function of EDB-FN is interesting. However, some points should be addressed before publication.
<Major comments>
- In the image of the MDA-MB-468-TGFbeta cells shown in figure 2A, cells were too dense to investigate cell morphology. The authors should reduce the confluency.
- Various mesenchymal markers have been identified in breast cancer cells. In figure 2DE, the authors should consider investigating the expression of other markers, such as vimentin, ZEB1, ZEB2 and Snail.
- The authors used TGFbeta-treated and chemoresistant cells. Data shown in figure 3EF suggest that the expression level of EDB-FN is various in derivative cell lines and there are at least two populations. However, the manuscript did not describe whether the authors isolated single cell clones or not.
- Are the conditions of figure 2B and 4G the same? The phenotypes look different.
- In figure 5A, it is unclear whether total fibronectin level was changed by gene knockdown or not. Therefore, it is also unclear whether SRp55 alters fibronectin splicing to generate EDB-FN or SRp55 regulates total fibronectin expression. The authors should show total fibronectin expression in figure 5A. If total fibronectin level is reduced by gene knockdown, EDB-FN level normalized by total fibronectin level should be shown.
<Minor comments>
- Knockdown experiments for EDB-FN and SRSF6 in MDA-MB-231 and/or Hs578T cells are interesting.
- Did the authors check SRSF6 in TCGA and GTEx databases?
Author Response
Reviewer 2.
Vaidya et al. found that upregulation of EDB-FN in aggressive breast cancers. The authors used MCF-7 and MDA-MB-468 cells. TGFbeta-treated cells and chemoresistant cells showed higher EDB-FN expression than parental cells. Moreover, in these cells, AKT inhibitor treatment downregulated EDB-FN expression and reduced the invasive ability. SRSF6 knockdown also reduced EDB-FN expression and the invasive ability. The manuscript is well organized. The function of EDB-FN is interesting. However, some points should be addressed before publication.
<Major comments>
In the image of the MDA-MB-468-TGFbeta cells shown in figure 2A, cells were too dense to investigate cell morphology. The authors should reduce the confluency.
R: The MDA-MB-468-TGFbeta cells look too dense because they proliferate faster. The images of all 3 MDA-MB-468 cell line types were taken at the same time post-plating for accurate comparison. However, the insert included with the main image depicts the cell boundaries more clearly.
Various mesenchymal markers have been identified in breast cancer cells. In figure 2DE, the authors should consider investigating the expression of other markers, such as vimentin, ZEB1, ZEB2 and Snail.
R: The reviewer has a good point. However, because we do not want to make the paper about EMT markers, we did not elaborate on all of them. Multiple pathways like Hedgehog, NF-κB , etc. could be involved (as reviewed by Delou et al., Highlights in Resistance Mechanism Pathways for Combination Therapy. Cells. 2019 Sept 8(9):1013). There already have been so many reports about different mesenchymal markers in breast cancer. Emphasis on these markers would complicate the interpretation and take away from the focus of EDB-FN and invasion. Nevertheless, we will perform analysis of other markers in our future studies.
The authors used TGFbeta-treated and chemoresistant cells. Data shown in figure 3EF suggest that the expression level of EDB-FN is various in derivative cell lines and there are at least two populations. However, the manuscript did not describe whether the authors isolated single cell clones or not.
R: This is an interesting observation. We did not single out the clones. We looked at the whole population. It is more likely that since the dots in 3E-F represent technical replicates from independent experiments, the 2-ddCt values vary from experiment-to-experiment, and that is what we observe in the PCR data. But, given our western blot results, we cannot deny the possibility of a mix of cells that express different markers and work in conjunction with each other. In fact, cancer cells are shown to form multiple zones during the migratory process, where leader cells are mesenchymal-like while the follower cells can retain their epithelial features (Leonard & Godin, Translational Cancer Research. 2017 Vol 6; Konen et al., Nat Comm 2017. 8:15078).
Are the conditions of figure 2B and 4G the same? The phenotypes look different.
R: This is also a great obervation. The conditions are the same. The cells are very sensitive when it comes to 3D culture, number of cells plated, number of days elapsed, etc. So while they may form spheroids early on, culturing a higher number of cells or using higher passage cells can also lead to the formation of proliferative networks, as in the case of the invasive MCF populations.
In figure 5A, it is unclear whether total fibronectin level was changed by gene knockdown or not. Therefore, it is also unclear whether SRp55 alters fibronectin splicing to generate EDB-FN or SRp55 regulates total fibronectin expression. The authors should show total fibronectin expression in figure 5A. If total fibronectin level is reduced by gene knockdown, EDB-FN level normalized by total fibronectin level should be shown.
R: It is most likely that total fibronectin level will also reduce, probably indirectly, given that SRp55 controls various oncogenic signaling pathways by alternative splicing of proteins like FGFR1 (Ghigna et al., Curr Genomics 2008, 9(8)556-70). Given that FN1 is also expressed in normal tissues, it is not a suitable marker. Hence we focussed only on EDB-FN. Since the G4 antibody specifically binds to the EDB domain of FN1, the silencing observed is only for EDB-FN. To study the mechanism of splicing, in-depth studies analyzing primary and processed transcripts will be conducted in the future.
<Minor comments>
Knockdown experiments for EDB-FN and SRSF6 in MDA-MB-231 and/or Hs578T cells are interesting.
R: This is a good point. We selected the relatively low EDB-FN-expressing MCF7 and MDA-MB-468 cell lines. But for our future studies, we will perform knockdown of EDB-FN in the EDB-FN-rich MDA-MB-231 and Hs578T cells lines to evaluate its effect on these invasive lines.
Did the authors check SRSF6 in TCGA and GTEx databases?
R: This is a great point. We did look at the expression of SRSF6 in TCGA and GTEx databases. Although we did not see a significant difference in level between tumor
vsnormal, interestingly, we found a significant positive correlation between EDB-FN and SRSF6 expression in BRCA tumors (*p=7.7e-05) but not in normal breast mammary tissue (p=0.77), as shown below. It will be interesting to pursue this correlation in various cell types in our future studies.

Round 2
Reviewer 1 Report
Regarding EDB specificity and EMT problems, I suggest that the authors detail what they responded to my comments, including references they cited, in the Discussion section, where they should also depict certain important issues and designed experiments (per our communication) to be explored in the future.
Author Response
Comment: Regarding EDB specificity and EMT problems, I suggest that the authors detail what they responded to my comments, including references they cited, in the Discussion section, where they should also depict certain important issues and designed experiments (per our communication) to be explored in the future.
Response: We thank the reviewer for this helpful comment. As suggested, we have added the following text with additional citations in the Discussion:
Lines 436-442: While we observed increased expression of the pro-invasive protein Slug in both the cell lines with long-term TGF-β treatment and drug resistance, it is likely that each of these cell lines acquired their survival advantages through other known pathways like Hedgehog, NF-κB, PI3K-AKT-mTOR [47], and additional here-to-fore unstudied mechanisms. Although pAKT signaling was robustly upregulated in the cells, it would be of significant future interest to gain in-depth insight into the precise causal factors of their increased motility.
Lines 446-454: Given that FN1 has several other alternatively spliced isoforms, including EDB/EDA- FN1, and EDA-FN, further extensive studies are required to evaluate the specificity and/or sufficiency of the role of EDB-FN in invasion. In view of the numerous studies demonstrating overexpression of EDB/EDA- FN1 in tumorigenic ECM [11,48], it can be speculated that it may also be upregulated in our TGF-β-treated and drug-resistant breast cancer cells. However, its abundance in healthy tissues renders it an unsuitable tumor-specific marker, and did not comprise the focus of our research. On the other hand, the oncofetal isoform EDA-FN is under investigation as a potential oncomarker, and its role in breast cancer is described elsewhere [49,50].
Reviewer 2 Report
No comment
Author Response
The reviewer did not ask for further revision.
This manuscript is a resubmission of an earlier submission. The following is a list of the peer review reports and author responses from that submission.
Round 1
Reviewer 1 Report
This manuscript entitled "Extradomain-B Fibronectin is a Molecular Marker of Invasive Breast Cancer Cells" is attempted to provide evidence that overexpression of EDB-FN in invasive breast cancer cells potentially can serve as a molecular marker for breast cancer patients' prognosis. Overall, such intention would ultimately benefit breast cancer patients if the authors are able to further provide stronger supporting results to complement current evidence which is insufficiently convincing yet. It looks to me that some of the results are not necessary and dispensable to the conclusion made in the title.
Specific critiques:
1. While it is a novel thought that EDB-FN serves a prognostic marker for the invasive phenotype or malignancy of breast cancer patients, the comparison between EDB-FN expression in normal breast tissues and breast tumor tissues is logically barking the wrong tree. A logically correct comparison should be between EDB-FN expression in non-invasive (or less malignant) and highly invasive (or malignant) breast tumor tissues. Alternatively, a series of different pathological stages can be used to associate the EDB-FN expression.
2. Tumor cells may not be the only cell type capable of expressing EDB-FN in the entire tumor tissues. Detection of tumor tissue-derived mRNAs does not necessarily warrant the expression levels in tumor cells per se, unless in situ FISH or even better off IHC (detection the protein levels by EDB-FN-specific Abs or the ZD2-Cy5.5 EDB-FN-binding fragment) technique is performed to directly visualize the expression in tumor cells of the patients' tumor tissues.
3. The distributions, locations, and functions of all FN molecules within tumor tissues are very complex and difficult to draw solid conclusions in terms of cancer diagnosis and prognosis. Therefore, if carefully evidenced, EDB-FN may exert its specific role for those purposes. To corroborate the specific association of EDB-FN with invsiveness of breast cancer cells, addition of a control of EDB-free FN expression should be employed to highlight the specificity of EDB-FN as a potent prognostic marker for breast cancer patients.
4. In breast cancer cell lines, non-invasive versus invasive lines (but not two non-invasive lines, namely MCF7 and MDA-MB-468) should first be used in the 3D invasion assays and EDB-FN binding assays (in addition to the dection of mRNA levels) to demonstrate a strong correlation between general breast cancer cell invasivenesss and EDB-FN-specific expression (EDB-FN as a marker for cancer patients' tumor invasiveness) before the comparison between naive non-inivasive/AKT 291 inhibitor MK2206-HCl-treated and TGF-b-treated or chemoresistant breast cancer cell lines, which is to answer a distinct question, namely whether EDB-FN expression is associated with TGF-b-/chemoresistance-triggered and MK2206-HCl-suppressed tumor invasiveness and able to serve as a therapy-predictive marker.
5. Since EMT- or partial EMT-related issues concerning tumor cell migration and invasion are highly controversial and debatable and the EMT-related results so-called the hybrid E-M phenotype (or partial EMT) in Figures 2. 3, and 5 did not exhibit a well-defined pattern, I don't see any necessity or indispensable role for the purpose of reinforcing EDB-FN as a prognostic marker for cancer invasiveness in breast cancer patients. Therefore, I suggest that the EMT-related issues are removed from this manuscript.
6. In Figure 5B and 5K (5I should be removed and thus not included), labeling seems incorrect as the comparisons should be MCF7/468-TGFb or MCF7/468-DR in the absence or presence of MK2206, but not MCF7/468 without MK2206 and MCF7/468-TGFb/DR with MK2206.
7. In Figure 5, MK2206 is a strong AKT inhibitor capable of dephosphorylating pAKT-T308 and -S473. Importantly, AKT is a well known survival factor. It is very likely that MK2206 induced apoptosis of MCF7 and 468 when AKT was inactivated. If that was so, all the effects triggered by MK2206 may just be a consequence of cell death and had nothing to do with the reduced invasiveness, EDB-FN expression, and malignancy, unless cell viability was tested and no cell death was manifested.
8. In Figure 5, were western results derived from 2D or 3D culture or both cultures made no difference? A discussion should be elaborated regarding the differences for the two culturing conditions.
Author Response
We thank the reviewers for their critical review of the manuscript and valuable comments. We have addressed the comments in the manuscript (and point-wise below) and we believe that the suggested changes have helped to strengthen the quality of our manuscript.
Reviewer 1
This manuscript entitled "Extradomain-B Fibronectin is a Molecular Marker of Invasive Breast Cancer Cells" is attempted to provide evidence that overexpression of EDB-FN in invasive breast cancer cells potentially can serve as a molecular marker for breast cancer patients' prognosis. Overall, such intention would ultimately benefit breast cancer patients if the authors are able to further provide stronger supporting results to complement current evidence which is insufficiently convincing yet. It looks to me that some of the results are not necessary and dispensable to the conclusion made in the title.
We are grateful for your positive feedback and encouraging outlook on the research.
Specific critiques:
While it is a novel thought that EDB-FN serves a prognostic marker for the invasive phenotype or malignancy of breast cancer patients, the comparison between EDB-FN expression in normal breast issues and breast tumor tissues is logically barking the wrong tree. A logically correct comparison should be between EDB-FN expression in non-invasive (or less malignant) and highly invasive (or malignant) breast tumor tissues. Alternatively, a series of different pathological stages can be used to associate the EDB-FN expression.We concur that it is important to compare EDB-FN expression between different subtypes and stages of breast tumor tissues to establish it as a prognostic marker of invasive breast cancer. In our previous studies, we have indeed done similar experiments, comparing in vivo EDB-FN expression between less invasive MCF7 and highly invasive MDA-MB-231, BT549, and Hs578T tumors (Han et al, Nat Comm 2017). In this study, we presented the comparison of EDB-FN levels between normal breast tissue and breast tumor tissue as a whole, and then focused mainly on to comparing breast cancer cells of different degrees of invasiveness, since the scope of this manuscript is limited to cells.
Tumor cells may not be the only cell type capable of expressing EDB-FN in the entire tumor tissues. Detection of tumor tissue-derived mRNAs does not necessarily warrant the expression levels in tumor cells per se, unless in situ FISH or even better off IHC (detection the protein levels by EDB-FN-specific Abs or the ZD2-Cy5.5 EDB-FN binding fragment) technique is performed to directly visualize the expression in tumor cells of the patients' tumor tissues.The reviewer brings up a very significant point. Since tumors are a heterogenous mix of different types of cells, an accurate estimation of EDB-FN secretion would require IHC and/or in situ staining of tissue samples. We are indeed in the process of acquiring patient tissue samples for detecting in vivo EDB-FN expression for future study, which will build up on this in vitro cell study.
The distributions, locations, and functions of all FN molecules within tumor tissues are very complex and difficult to draw solid conclusions in terms of cancer diagnosis and prognosis. Therefore, if carefully evidenced, EDB-FN may exert its specific role for those purposes. To corroborate the specific association of EDB-FN with invasiveness of breast cancer cells, addition of a control of EDB-free FN expression should be employed to highlight the specificity of EDB-FN as a potent prognostic marker for breast cancer patients.As the reviewer has rightly mentioned, the function of EDB-FN is much more complex and is likely to be temporally and spatially regulated. Teasing apart the functions of EDB-FN and EDB-free FN (with the added complexity of the presence of EDA and other alternative splicing variants) is a highly interesting topic. However, it will require a comprehensive research design and will be the focus of our future studies and has been added in the Discussion.
In breast cancer cell lines, non-invasive versus invasive lines (but not two non-invasive lines, namely MCF7 and MDA-MB-468) should first be used in the 3D invasion assays and EDB-FN binding assays (in addition to the detection of mRNA levels) to demonstrate a strong correlation between general breast cancer cell invasivenesss and EDB-FN-specific expression (EDB-FN as a marker for cancer patients' tumor invasiveness)before the comparison between naive non-inivasive/AKT 291 inhibitor MK2206-HCl-treated and TGF-b-treated or chemoresistant breast cancer cell lines, which is to answer a distinct question, namely whether EDB-FN expression is associated with TGF-b-/chemoresistance-triggered and MK2206-HCl-suppressed tumor invasiveness and able to serve as a therapy-predictive marker.This is a great suggestion. We have included an additional figure (Fig 1A) which demonstrates a strong correlation between the invasiveness and EDB-FN-specific ZD2-Cy5.5 binding in all the 5 breast cancer cells lines listed in Fig 1B.
Since EMT- or partial EMT-related issues concerning tumor cell migration and invasion are highly controversial and debatable and the EMT-related results so-called the hybrid E-M phenotype (or partial EMT) in Figures 2. 3, and 5 did not exhibit a well-defined pattern, I don't see any necessity or indispensable role for the purpose of reinforcing EDB-FN as a prognostic marker for cancer invasiveness in breast cancer patients. Therefore, I suggest that the EMT-related issues are removed from this manuscript.Although this comment is reasonable, we believe that we should honestly report our observations. Reviewer 2 has suggested that these results be discussed more. The increased E-cadherin levels are very reproducible in our hands, and do suggest that the cells may have adapted a more plastic phenotype. However, since the major focus of this manuscript is to test for EDB-FN expression in the context of invasion, we have stressed it as such, and we have refrained from stating that EDB-FN could be a marker of E-M phenotype (section 3.3 and 3.4).
In Figure 5B and 5K (5I should be removed and thus not included), labeling seems incorrect as the comparisons should be MCF7/468-TGFb or MCF7/468-DR in the absence or presence of MK2206, but not MCF7/468 without MK2206 and MCF7/468-TGFb/DR with MK2206.This was a labeling mistake on our part. We have corrected the labeling and removed Fig. 5I.
In Figure 5, MK2206 is a strong AKT inhibitor capable of dephosphorylating pAKT-T308 and -S473. Importantly, AKT is a well known survival factor. It is very likely that MK2206 induced apoptosis of MCF7 and 468 when AKT was inactivated. If that was so, all the effects triggered by MK2206 may just be a consequence of cell death and had nothing to do with the reduced invasiveness, EDB-FN expression, and malignancy, unless cell viability was tested and no cell death was manifested.This is a very good point. Indeed, we observed significant cell death with MK2206 doses greater than 4 µM for the both the cell lines. Consequently, we reduced the dose to 2 µM for the less invasive MCF7 and 4 µM for the more invasive MDA-MB-468 cell derivatives, respectively. However, all the post-treatment experiments were set up with normalization controls in such a manner that the number of cells for DMSO-control and MK-treated cells was always equal. For instance, following MK2206 treatment, the cells were harvested and counted (we did not see significant cell death). After counting, an equal number of cells was plated on 3D Matrigel and in transwells. The total RNA and protein extracts harvested were also measured for concentration and equivalent amounts (1 µg and 40 µg) were used for RT-PCR and western blot, respectively, and then normalized again using actin as loading control.
We have added additional text for clarification in the Materials and Methods section.
In Figure 5, were western results derived from 2D or 3D culture or both cultures made no difference? A discussion should be elaborated regarding the differences for the two culturing conditions.This is a very good question. The protein extracts were derived from 2D culture. Although we also attempted to derive protein extracts from 3D culture, we found that the Recovery solution recommended for use with Matrigel culture degrades the matrix and consequently the fibronectin in it, adding an unpredictable variable for EDB-FN levels in our experiments. For this reason and to keep the protocol consistent for all proteins, we used 2D extracts for western blot. As the reviewer suggests, it would be interesting to see the protein expression changes in the context of 3D culture. As suggested in comment 1 and 2, using tumor tissue samples from patients is the right step in moving forward and we will determine the levels of AKT and splicing proteins in these samples to correlate them with EDB-FN expression and invasiveness in patients.
Nevertheless, we have added a clarification in the Materials and Methods section and additional text in the Discussion.
Reviewer 2 Report
The aim of the manuscript submitted by Vaidya et al. is to demonstrate that EDB-Fibronectin constitutes a reliable biomarker of aggressive and invasive breast cancers. Thus, they revealed that EDB-FN mRNA expression is higher in breast tumor than in normal tissue as well as in HR- compared to HR+ breast cancer cell lines. Then, they observed a correlation between the invasiveness properties of the cell lines studied and the EDB-FN expression. Finally, they studied the involvement of the AKT-SRp55 signalling pathway in the regulation of the EDB-FN expression. They concluded that EDB-FN could be a promising molecular marker for monitoring the progression of breast cancer.
Major points
I have some concerns about the interpretation of the results.
Statistical analyses have been performed using unpaired t-test all the long the manuscript. But, except in Fig 1A where this test is right, in Fig 1B a Kruskas Wallis test has to be performed instead of t-test and for the others analyses, given the small size of the samples (n=3), a U-Mann and Whitney test has to be used. Results have to be re-analysed with the good statistical tests Histograms of blot quantification have to be shown. Moreover, I have a concern with the western blot loading controls. Indeed, bands of actin are strictly the same for western blots showed in Fig 5A and 5J and, Fig 5B and 5I and K. Authors must clarify this point and provide the original images of the various western blots (complete membrane pictures of all blots: markers, protein studied and corresponding loading control for each blot). Fig 3 D-E: The authors write that in « both MCF-TGFβ and MCF7-DR cells showed upregulated N-cad and Slug expression while the E-cad expression did not significantly decrease (Figure 3D) ». I agree that E-cad did not significantly decrease… but it seems that, in contrast, it increases ! Later, they write that « MDA-MB-468-DR cells showed increased N-cad and Slug expression, compared to the parent cells (Figure 3E) » However, the bands corresponding to slug are almost undetectable, so the conclusion should be more cautious. To note that in MDA-MB-468-DR E-cad expression is also increased like in MCF7… This should be discussed. Fig 3F : The transwell experiments are performed for 2 days, which is a very long period of time for this test. How is it justified? In these experimental conditions, how the authors are sure that the invasion increase observed with MDA-MB-468-TGFbeta is not due to a TGFbeta treatment effect on cell proliferation as mentioned p 6 line 203 ? Fig 4C : Hoechst staining should be shown. Validation of siEDB by western blot should be added. This study is essentially correlative, it would be interesting to test the effects of siEDB on invasion and drug response of MCF7 and MDA-MD-468 cell lines.
Minor point
There is a mistake in the Fig 2 legend. In the legend, A and B correspond to the type of cell lines while in the Figure they correspond to the culture condition.
Author Response
We thank the reviewers for their critical review of the manuscript and valuable comments. We have addressed the comments in the manuscript (and point-wise below) and we believe that the suggested changes have helped to strengthen the quality of our manuscript.
Reviewer 2
The aim of the manuscript submitted by Vaidya et al. is to demonstrate that EDB-Fibronectin constitutes a reliable biomarker of aggressive and invasive breast cancers. Thus, they revealed that EDB-FN mRNA expression is higher in breast tumor than in normal tissue as well as in HR- compared to HR+ breast cancer cell lines. Then, they observed a correlation between the invasiveness properties of the cell lines studied and the EDB-FN expression. Finally, they studied the involvement of the AKT-SRp55 signalling pathway in the regulation of the EDB-FN expression. They concluded that EDB-FN could be a promising molecular marker for monitoring the progression of breast cancer.
Major points
I have some concerns about the interpretation of the results. Statistical analyses have been performed using unpaired t-test all the long the manuscript. But, except in Fig 1A where this test is right, in Fig 1B a Kruskas Wallis test has to be performed instead of t-test and for the others analyses, given the small size of the samples (n=3), a UMann and Whitney test has to be used. Results have to be re-analysed with the good statistical tests.We have reanalyzed Fig 1A (now Fig 1C) using the Kruskal Wallis test, and the significant p value still holds true. The other groups were re-analyzed using the Mann-Whitney test and the changes in significance (if any) are noted in the relevant figure legends. We have also incorporated the new tests in the Materials and Methods section.
Histograms of blot quantification have to be shown. Moreover, I have a concern with the western blot loading controls. Indeed, bands of actin are strictly the same for western blots showed in Fig 5A and 5J and, Fig 5B and 5I and K. Authors must clarify this point and provide the original images of the various western blots (complete membrane pictures of all blots: markers, protein studied and corresponding loading control for each blot).We understand the concern raised by the reviewer and it is a good point. We used the same actin blot because these images were derived from the same samples. Because a lot of the proteins detected are around the same molecular weight as actin (45 kD), it is not physically possible to run actin loading control for each blot of same samples. To avoid confusion, we have compiled the proteins together in Fig 5. Nevertheless, we have also included uncropped blots for all the proteins in a separate attachment.
Fig 3 D-E: The authors write that in « both MCF-TGFβ and MCF7-DR cells showed upregulated N-cad and Slug expression while the E-cad expression did not significantly decrease (Figure 3D) ». I agree that E-cad did not significantly decrease… but it seems that, in contrast, it increases! Later, they write that « MDA-MB-468-DR cells showed increased N-cad and Slug expression, compared to the parent cells (Figure 3E) » However, the bands corresponding to slug are almost undetectable, so the conclusion should be more cautious. To note that in MDA-MB-468-DR E-cad expression is also increased like in MCF7… This should be discussed.We concur with this comment. We have revised the text to describe the E-cad, N-cad and Slug expression changes in each of the 4 invasive cell derivatives. The western blot image for Slug has been replaced with another independent replicate for a better representation of the result. We have added additional text to note the observation that E-Cad increases in 468-DR cells. We think this is likely indicative of high cancer cell plasticity, whereby the cells can rapidly regulate their E-cad and N-cad levels, based on the spatial and temporal context.
Fig 3F : The transwell experiments are performed for 2 days, which is a very long period of time for this test. How is it justified? In these experimental conditions, how the authors are sure that the invasion increase observed with MDA-MB-468-TGFbeta is not due to a TGFbeta treatment effect on cell proliferation as mentioned p 6 line 203?
This is a good question. We repeated the transwell experiment with a shorter time point (1 day) to compare MDA-MB-468 and MDA-MB-468-TGF-β cell invasion and we also observed increased migration for the latter. Briefly, 200,000 cells were plated in serum-free starving media in the matrigel-coated inserts and stained with crystal violet after 1 day. Ideally, it would be help to use a proliferation marker like Mitomycin C to tease apart the proliferative and invasive characteristics. However, in the limited time frame given for revisions, we reduced the incubation time with starving conditions to determine the invasion of TGF-β-treated cells.
Fig 4C : Hoechst staining should be shown. Validation of siEDB by western blot should be added.
We added the composite image of Hoechst staining and ZD2-Cy5.5 in Fig 4E. In addition, for validation of EDB-FN knockdown, we also added RT-PCR figure. Because EDB-FN is a secreted protein, the mRNA levels coupled with ZD2-Cy5.5 binding assay in live culture would be a better indicator of EDB-FN silencing compared to harvesting the extracts for western blot.
This study is essentially correlative, it would be interesting to test the effects of siEDB on invasion and drug response of MCF7 and MDA-MB-468 cell lines.This is a great suggestion. However, it will require a comprehensive research design, comprising EDB-free FN as suggested by Reviewer 1, and will be the focus of our future studies.
Minor pointThere is a mistake in the Fig 2 legend. In the legend, A and B correspond to the type of cell lines while in the Figure they correspond to the culture condition.
This was a proofreading mistake on our part. We have corrected the legend to separate 2D and 3D culture conditions.
Round 2
Reviewer 1 Report
Although the manuscript is now somewhat improved upon revision, some old and new problems remain unresolved.
It’s good that Fig. 1A has been added for the correlation between EDB-FN expression and tumor cell invasiveness. However, I don’t see too many cells as visualized by the Hoechst dye staining. A fair comparison should be based on the same cell numbers or density. If 3D culture is not possible to equivalent cell density, 2D culture can alternatively be used, as all the IB results were derived from 2D culture. It’s weird that the nuclear staining was particularly weaker than other cell lines in Fig. 1A. Evaluation of EDB-FN expression in clinical samples firmly enhances the significance of conclusion from data resulted from cell cultures. Therefore, it is important to test whether EDB-FN expression is indeed higher in invasive or more malignant types of breast cancer cells than in non-invasive or less malignant ones in human samples. Comparing EDB-FN expression in human normal and breast cancer tissues (as a whole) is illogical and meaningless. I fail to see a good reason in authors’ response to keep Fig. 1C. The authors should either revise the clinical results accordingly or just simply remove Fig. 1C. However, if they choose to remove the clinical data, the scientific soundness will be depreciated. Without demonstrating the specificity of EDB-FN expression in invasive breast cancer cells, the title “Extradomain-B Fibronectin is a Molecular Marker of Invasive Breast Cancer Cells” is insufficiently supported by current results in the manuscript. It could well be that invasive breast cancer cells only increased general FN expression. Such possibility would significantly depreciate the value of EDB-FN as a specific marker in invasive breast cancer cells. Regarding hybrid E-M phenotype issues, in order to publish a manuscript in a high prestige Journal such as Cells, it is not just a matter of honestly reporting observations irrelevant to the main theme but presenting strongly supporting results. It was very inconsistent as to the expression of E-M markers in Fig. 3. For example, whereas mRNA levels of E-cadherin (E-cad) in MCF7-TGFb and MCF-DR were significantly reduced, the protein level of E-cad in MCF-DR were obviously increased. Furthermore, unlike MCF7, mRNA levels of E-cad were unchanged in MDA-MB-468, MDA-MB-468-TGFb, and MDA-MB-468-DR, whereas the protein level of E-cad in MDA-MB-468-DR was clearly increased. I could not find any rule of definition for such messy relationships. E-cad is very well-known for its inhibitor effect on cancer cell migration/invasion. Since the migration and invasion abilities of both MCF7-TGFb/DR and MDA-MB-468-TGFb/DR were increased, increase of E-cad protein expression in these cells is simply not in line with the elevated migration/invasiveness. Therefore, I stay with my last suggestion.Reviewer 2 Report
The authors have replied to most questions raised.
1) However, there is still a concern about western blot loading controls.
As required, original western blots have been provided. Although it is difficult to find one's way through all the data, I have a serious concern about the actin blot of the Figure 3 and figure 5 shown in supplemental data. Indeed, it seems that it is the same blot more or less exposed for both series of western blots BUT the lines do not correspond to the same conditions: Figure 3 lines : MCF7, MCF7-TGFb, 468 and 468-TGFbeta become in Figure 5: 468, 468-TGFbeta, 468 and 468-DR. How can the authors explain this?
In addition, the actin used as a loading control must be revealed on the same membrane as the proteins of interest even after stripping if necessary.
2) For the transwell experiments: the new experiments performed with a shorter period of time than previously have to be shown instead of those of the first version of the manuscript. Ideed, materials and methods section and figures 3 and 5 have not been changed in the revised version.